# Randomized single oral dose phase 1 study of safety, tolerability, and pharmacokinetics of Iminosugar UV-4 Hydrochloride (UV-4B) in healthy subjects

Michael Callahan[1], Anthony M. Treston[2¤a], Grace Lin[2¤b], Marla Smith[2¤c], Brian Kaufman[3], Mansoora Khaliq[2¤d], Lisa Evans DeWald[2], Kevin Spurgers[2], Kelly L. Warfield[2]*, Preeya Lowe[2], Matthew Duchars[2¤e], Aruna Sampath[2¤f], Urban Ramstedt[3]

1 Division of Infectious Diseases, Massachusetts General Hospital, Massachusetts, United States of America, 2 Emergent BioSolutions Inc, Gaithersburg, Maryland, United States of America, 3 AbViro, Bethesda, Maryland, United States of America

¤a Current address: Treadwell Therapeutics, Toronto, Canada
¤b Current address: Esperion Therapeutics, Inc, Ann Arbor, Michigan, United States of America
¤c Current address: PACTS CTO, Washington, District of Columbia, United States of America
¤d Current address: GSK, Rockville, Maryland, United States of America
¤e Current address: The Vaccines Manufacturing and Innovation Center, Oxford, United Kingdom
¤f Current address: Dynavax Technologies Corporation, Emeryville, California, United States of America
* warfieldK@ebsi.com

**Data Availability Statement:** All relevant data are within the manuscript and its Supporting Information files. Supplemental Information-3

## Abstract

### Background

UV-4 (*N*-(9'-methoxynonyl)-1-deoxynojirimycin, also called M*O*N-DNJ) is an iminosugar small-molecule oral drug candidate with *in vitro* antiviral activity against diverse viruses including dengue, influenza, and filoviruses and demonstrated *in vivo* efficacy against both dengue and influenza viruses. The antiviral mechanism of action of UV-4 is through inhibition of the host endoplasmic reticulum-resident α-glucosidase 1 and α-glucosidase 2 enzymes. This inhibition prevents proper glycan processing and folding of virus glycoproteins, thereby impacting virus assembly, secretion, and the fitness of nascent virions.

### Methodology/Principal findings

Here we report a first-in-human, single ascending dose Phase 1a study to evaluate the safety, tolerability, and pharmacokinetics of UV-4 hydrochloride (UV-4B) in healthy subjects (ClinicalTrials.gov Identifier NCT02061358). Sixty-four subjects received single oral doses of UV-4 as the hydrochloride salt equivalent to 3, 10, 30, 90, 180, 360, 720, or 1000 mg of UV-4 (6 subjects per cohort), or placebo (2 subjects per cohort). Single doses of UV-4 hydrochloride were well tolerated with no serious adverse events or dose-dependent increases in adverse events observed. Clinical laboratory results, vital signs, and physical examination data did not reveal any safety signals. Dose-limiting toxicity was not observed; the maximum tolerated dose of UV-4 hydrochloride in humans has not yet been determined (>1000 mg). UV-4 was rapidly absorbed and distributed after dosing with the oral solution

includes the values behind the means, standard deviations and other measures reported, including the values used to build graphs. Per standard practice, raw data (LC-MS/MS mass chromatogram areas for samples and standard curves) have been processed to generate concentration data which are provided as Tables.

**Funding:** This research was funded in part by the National Institute of Allergy and Infectious Diseases (NIAID; https://www.niaid.nih.gov/), National Institutes of Health, Department of Health and Human Services, under contract number HHS272201100030C (awarded to U.R). The funders assisted with study design and monitoring, and data analysis, but not with decision to publish or preparation of the manuscript. Authors A.M.T, M.S, B.K, M.K, K.L.W, P.L, M.D, A.S, and U.R received partial salary support from the indicated NIAID contract.

**Competing interests:** I have read the journal's policy and the authors of this manuscript have the following competing interests: L.DW, K.S, K.L.W, and P.L are employees of Emergent BioSolutions Inc. A.T and M.K carried out the work for this program on behalf of Emergent BioSolutions Inc. and the work is unrelated to current employers.

formulation used in this study. Median time to reach maximum plasma concentration ranged from 0.5–1 hour and appeared to be independent of dose. Exposure increased approximately in proportion with dose over the 333-fold dose range. UV-4 was quantifiable in pooled urine over the entire collection interval for all doses.

## Conclusions/Significance

UV-4 is a host-targeted broad-spectrum antiviral drug candidate. At doses in humans up to 1000 mg there were no serious adverse events reported and no subjects were withdrawn from the study due to treatment-emergent adverse events. These data suggest that therapeutically relevant drug levels of UV-4 can be safely administered to humans and support further clinical development of UV-4 hydrochloride or other candidate antivirals in the iminosugar class.

## Trial registration

ClinicalTrials.gov NCT02061358 https://clinicaltrials.gov/ct2/show/NCT02061358.

## Author summary

Dengue Fever, a disease caused by infection with dengue virus, is a significant global health problem. Some estimates indicate that ~390 million dengue virus infections occur each year, resulting in ~ 500,000 cases of severe dengue disease which can be fatal. One Dengue virus vaccine (Sanofi Pasteur's Dengvaxia) has been approved in some countries for people aged 9 to 45 years but has limited effectiveness and the potential for increased risk of severe dengue in seronegative individuals. There are no other drugs or vaccines widely available to prevent or treat dengue virus disease given the European Medicines Agency announced delay in approval decision on Takeda's TAK-003 candidate. Therefore, development of a drug to treat dengue is a global public health priority. This would be an important tool to go along with vaccination, potentially saving lives and relieving human suffering caused by dengue virus infections around the world. UV-4 is an antiviral drug candidate for dengue virus and additional acute viral diseases. An important attribute of UV-4 is that it does not target the virus directly. This has potential advantages such as preventing the virus from developing resistance to the drug. In previous studies, UV-4 or the hydrochloride salt UV-4B were well tolerated and ameliorated dengue and influenza virus disease in animals. Here, we report the first evaluation of UV-4 hydrochloride administration to healthy human volunteers. UV-4 as the hydrochloride was safe at all doses tested up to 1000 mg. These results support the further development of UV-4 for antiviral activity against dengue or other acute viral diseases.

## Introduction

Iminosugar molecules are sugar mimetics in which a carbon or nitrogen atom replaces the cyclic oxygen in the sugar ring [1]. Iminosugars act as competitive and reversible inhibitors of cellular enzymes which act on sugar substrates and can impact diverse cellular processes, and be administered for therapeutic benefit [2]. The iminosugar diabetes drugs acarbose and miglitol inhibit membrane-bound intestinal $\alpha$-glucosidase enzymes, inhibiting breakdown of

dietary carbohydrates into glucose [3,4]. The iminosugar chemical class has an established safety profile in humans with five products approved globally including miglustat (Zavesca) for the treatment of Gaucher disease and Niemann-Pick Type C [5]; migalastat (Galafold) for the treatment of Fabry disease [6]; and miglitol (Glyset), acarbose (Precose), and voglibose (Basen) for the treatment or prevention of type II diabetes mellitus [7].

Iminosugars are also being developed as antiviral therapeutics [2,8]. Diverse viral proteins are co-translationally glycosylated [9] in the host endoplasmic reticulum (ER) and rely on glycosylation for incorporation into virions, secretion, and function [10]. Iminosugars that target ER α-glucosidase I and II (host cell enzymes required for processing and proper folding of many viral glycoproteins) display *in vitro* antiviral activity against a broad range of enveloped viruses, including dengue virus (DENV), Ebola, influenza A, and hepatitis C viruses [11–15].

UV-4 is an iminosugar drug candidate intended for initial use as an antiviral agent for uncomplicated DENV infection. DENV is estimated to infect up to 390 million people worldwide each year [16]. Approximately 500,000 people with severe dengue require hospitalization annually; representing a major public health burden [17,18]. There is currently no antiviral agent available to treat dengue infections. There are several DENV vaccines in development [19,20], of which CYD-TDV (Dengvaxia, Sanofi) is approved in several dengue-endemic countries. However, current recommendations limit Dengvaxia use to patients who are dengue seropositive, due to limited effectiveness and safety concerns associated with antibody dependent enhancement [21–23]. Development of a therapeutic to treat DENV infections remains a global public health priority; such a drug would be an important tool to complement vaccination strategies.

UV-4 (*N*-(9'-methoxynonyl)-1-deoxynojirimycin) is closely related to the approved iminosugar miglustat. N-butyl-deoxynojirimycin UV-4 hydrochloride has potent *in vitro* activity against multiple isolates of all four DENV serotypes (DENV 1–4) [14]. The efficacy of UV-4 and UV-4 hydrochloride has also been characterized against DENV in mice using both direct infection (virus only) and an antibody-dependent enhancement (ADE; virus plus exogenous DENV-specific antibodies) model of severe dengue disease [24–26]. UV-4 protected mice from lethal DENV infection in a dose-dependent manner, reduced viral titer in tissues, and decreased cytokine levels in circulation [25]. UV-4 has proven broad-spectrum activity with efficacy ex vivo against H1N1, H3N2, and influenza B strains, and *in vivo* against lethal infection with H3N2, influenza B, and mouse-adapted oseltamivir -sensitive and -resistant influenza A (H1N1) strains [24,27]. Low potential for development of viral resistance has been confirmed with DENV and influenza [26,28]. Recent reports have identified potential of UV-4 and other iminosugars as therapeutics for coronaviruses [29,30] and inflammatory disorders associated with excess pathogen-induced cytokine secretion [31]. These findings supported evaluation of UV-4 in clinical trials.

Here, we present the results of a Phase 1a single ascending dose (SAD) study in which UV-4 was administered for the first time in humans. The study evaluated the safety, tolerability, and pharmacokinetics (PK) of increasing single oral doses of UV-4 hydrochloride in healthy subjects.

## Materials and methods

### Ethics statement

The clinical study protocol and other relevant study documents were reviewed and approved by the Midlands Independent Review Board (IRB) (Overland Park, KS) prior to subject screening and enrollment. The study was carried out in accordance with the Declaration of Helsinki, Office of Human Research Protections, Federalwide Assurance for the Protection of Human

Subjects, Good Clinical Practice (GCP) and local regulatory requirements [32]. All subjects provided written informed consent prior to study-related procedures.

## Study design

UV-4 ((2R,3R,4R,5S)-2-(hydroxymethyl)-1-(9-methoxynonyl)piperidine-3,4,5-triol, also called *N*-(9'-methoxynonyl)-1-deoxynojirimycin, or MON-DNJ) hydrochloride was evaluated in a first-in-human, randomized, single-center, double-blind, placebo-controlled, parallel group, single ascending dose Phase 1a clinical trial in healthy subjects (**Figs 1 and 2**, Clinical-Trials.gov Identifier NCT02061358 https://clinicaltrials.gov/ct2/show/NCT02061358). The primary objective was to determine the safety and tolerability of a single-ascending oral dose of UV-4 hydrochloride (**Fig 3**). Preparation of UV-4 as the hydrochloride salt confers higher solubility and high bioavailability in animal studies. The secondary objective was to determine PK parameters describing absorption and elimination of UV-4 following a single oral dose of UV-4 hydrochloride. Eight cohorts of eight subjects each were enrolled and completed the study. In each cohort, six subjects received UV-4 as the hydrochloride; two received placebo. Subjects were screened against study eligibility criteria up to four weeks (Days -28 to -2) before admission to the clinic (Day -1). Dosing was performed on an inpatient basis. Subjects remained in clinic for 48 hours post-dosing for PK sample collection and safety assessments

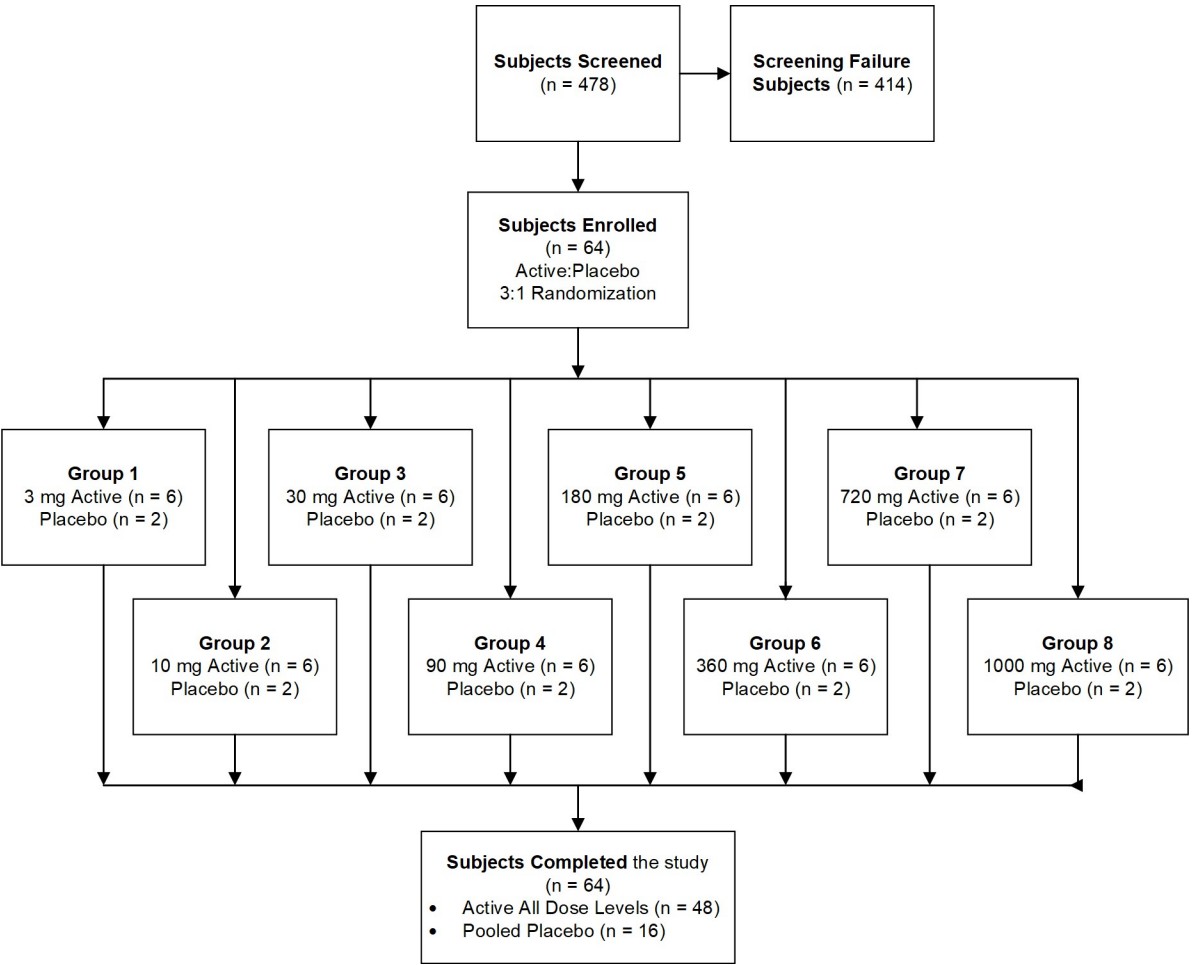

**Fig 1. Disposition of subjects.** N = number of subjects.

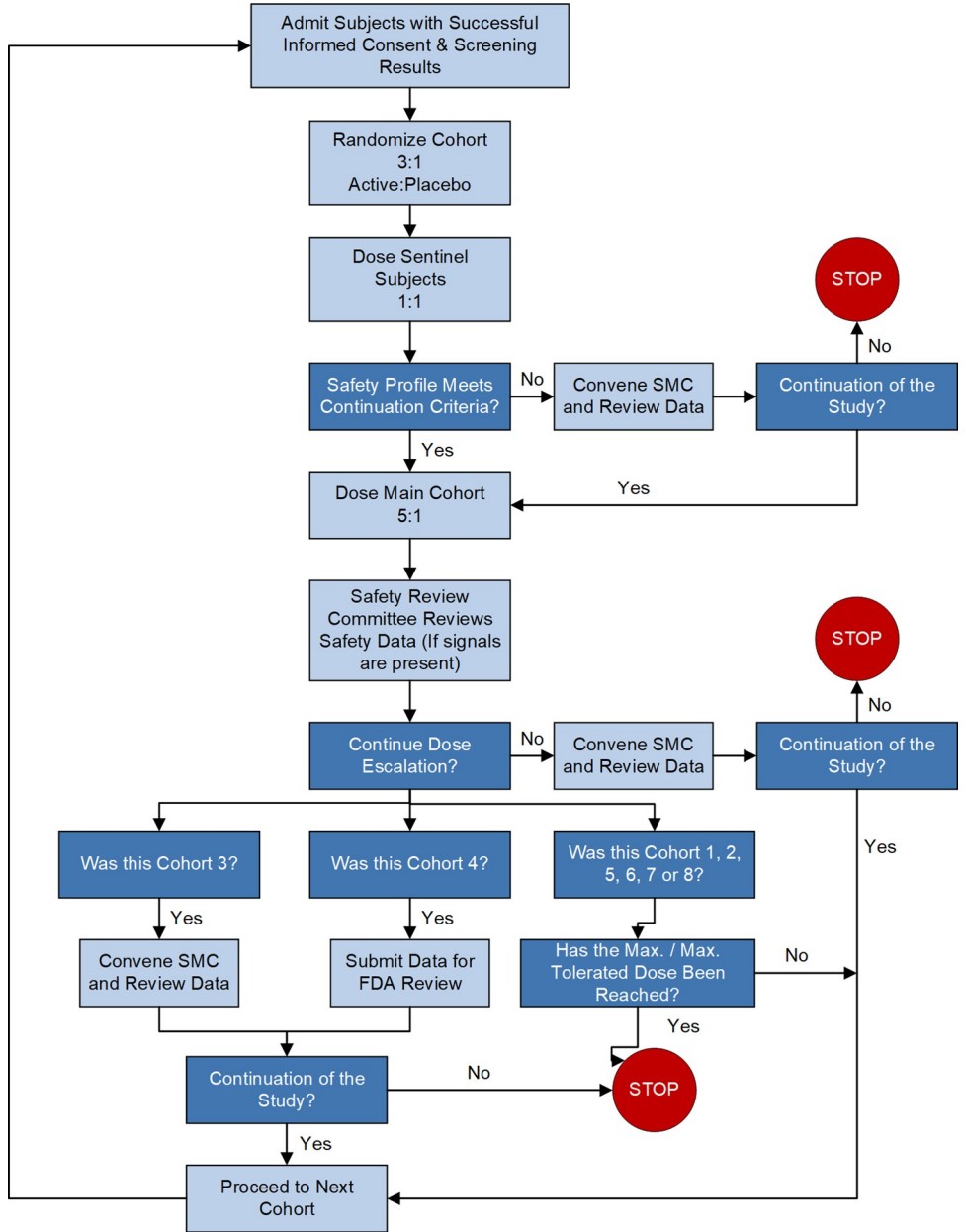

**Fig 2. Participants recruitment flowchart.** FDA = Food and Drug Administration; Max = maximum; SMC = Safety Monitoring Committee.

and were discharged from the clinic on Day 3. Subjects returned to the clinic on Day 4 for additional safety assessments, then again on Day 9 ± 1 for a final follow-up visit. The dose range was selected to meet regulatory (FDA) requirements for starting dose based on preclinical safety studies, with the upper doses predicted to give drug exposures in the range demonstrated to have antiviral therapeutic effect in nonclinical studies [24–26].

## Study subjects

Eligible subjects were healthy males and females, age 18–45 years, with a body mass index between 18 and 30 kg/m$^2$ inclusive and a minimum body weight of 60 kg. Subjects were

**Fig 3. Structural formula of UV-4 Hydrochloride.** Chemical structure of *N*-(9'-methoxynonyl)-1-deoxynojirimycin hydrochloride (UV-4B).

considered healthy based on medical history, physical exam, vital signs, 12-lead electrocardiogram (ECG), clinical laboratory tests (hematology, blood chemistry, urinalysis), viral serology, and drug and alcohol screening. Male and female subjects of childbearing potential were required to use adequate contraception over the course of the clinical trial. Subjects were excluded from study participation if they had a history of allergy to the iminosugar chemical class, creatinine clearance <90 mL/min (based on Cockcroft-Gault equation), proteinuria >30 mg/dL or abnormalities in physical examination suggestive of conditions that posed an increased risk to the subject (e.g., cardiovascular, pulmonary, endocrine, immune or other body system likely to pose a risk to the subject). Only subjects who met all inclusion and none of the exclusion criteria prior to administration of the investigational product were enrolled into the study (no exemptions were granted). A total of 64 subjects were enrolled (eight cohorts of eight subjects each). In each of the eight cohorts, six subjects were randomized to receive a single dose of UV-4 (3 to 1000 mg, depending on cohort) as the hydrochloride salt, and two subjects to receive placebo. Cohort size of six active subjects was based on analysis that increasing cohort size above six in a first-in-human study gains little in the ability to observe a drug induced event, and results in observing more events that are not related to study drug [33]. Two of the eight subjects in each cohort (sentinel subjects; one active and one placebo) received their doses at least 48 h before the remaining subjects in each cohort.

## Dosing

On each day of use, bulk drug substance (UV-4B for solubilization, SAI Life Sciences Ltd. Telangana, India) was removed from a single-use container, weighed at the study center, and solubilized in 20 mL sterile water for injection (SWFI) (Hospira Inc. Illinois, United States of America). Weight of solid drug substance was converted to dose of UV-4 by applying purity and salt correction factor per pharmacy manual when preparing the dosing solution. UV-4B solution was administered orally with 10 mL of taste-masking agent, OraSweet-SF (Paddock Laboratories LLC, Minneapolis, United States of America) in a total volume of 30 mL, to assist in maintaining study blinding. Placebo was SWFI (20 mL) plus OraSweet-SF (10 mL). Concentrations and doses were calculated and expressed as the active free base form, UV-4. This methodology is consistent with USP<1121>, which recommends that the strength of a drug product or preparation is expressed in terms of the active moiety. Following consumption of active or placebo solution, the dosing container was rinsed with 50 mL of tap water also consumed by subjects. The study pharmacist operated unblinded to prepare study drug and ensure active and placebo doses were administered to the appropriate randomized subject. Subjects were dosed after a 10-hour fast and continued to fast four hours post-dose until all

required post-dose assessments were completed. Consumption of water was not restricted, except for one hour prior to and two hours after administration of investigational product.

## Safety and tolerability assessment

Safety assessments included telemetry (pre-dose and 12 hours during and post-dosing), 12-lead ECG measurements, vital signs (supine and standing) including systolic blood pressure (SBP) and diastolic blood pressure (DBP) (mmHg), pulse rate (beats/minute), respiratory rate (breaths/minute), and oral temperature, physical examinations, clinical laboratory assessments (hematology, clinical chemistry, coagulation, urinalysis, platelet function), stool evaluation including occult blood, and adverse events (AEs).

Safety oversight was provided by an independent safety monitor (ISM), Safety Review Committee (SRC), and Safety Monitoring Committee (SMC). In addition, a cardiologist reviewed all cases with AEs related to heart rate and blood pressure. Summary statistics for subject baseline characteristics and measurements, disposition, exposure, and safety were tabulated by cohort in a blinded manner for the review by the SRC and SMC. Blinded individual case information was also supplied to the SMC and cardiologist for review. SRC reviews occurred after all subjects in a given cohort completed evaluations, or after predefined criteria triggered a review (halting criteria). Provided satisfactory safety review, subjects were enrolled into the next higher dose cohort. In addition, the SMC met to review all safety and exposure data after completion of Cohort 3. All AEs were coded to System Organ Class (SOC) and Preferred Term (PT) using the Medical Dictionary for Regulatory Activities (MedDRA), Version 16.1 [34].

All AEs were graded for severity and relationship to study product, according to one of the following categories:

- Grade 1: An event that is easily tolerated by the subject, causing minimal discomfort and not interfering with everyday activities;

- Grade 2: An event that is sufficiently discomforting to interfere with normal everyday activities;

- Grade 3: An event that prevents normal everyday activities;

- Grade 4: An event that requires an emergency room visit or hospitalization

## Pharmacokinetic evaluation

Serial blood samples were collected at pre-dose (0 hour) and 0.5, 1, 1.5, 2, 2.5, 3, 4, 6, 9, 12, 18, 24, 36, and 48 hours post-dose for assessment of UV-4 concentrations in plasma. Pooled urine samples were collected at pre-dose (-12 to 0 hour) and from 0 to 6, 6 to 12, 12 to 24, and 24 to 48 hours post-dose for assessment of UV-4 concentrations in urine. Plasma and urine concentrations of UV-4 were determined by means of validated, sensitive, and specific high-performance liquid chromatography/tandem mass spectrometric (LC-MS/MS) assays. UV-4 plasma and urine concentrations and PK parameters were listed and summarized by dose, as appropriate. Only the 48 subjects who received active investigational product were included in the PK analysis population.

## Dose proportionality

Dose proportionality was assessed based on whether the 90% confidence interval (CI) for the estimate of regression coefficient in a power model lies entirely within (0.96, 1.04). This interval ensured that a doubling of dosage was associated with no more than a 20% deviation from

doubling of exposure over the approximately 333-fold dose range (i.e., 3 mg to 1000 mg range) planned for this study. Simple least square (LS) method was used to estimate the parameters in the power model with $\log(C_{max})$ or $\log(AUC_{0\text{-inf}})$ against $\log(\text{dose})$.

## Results

The objective of this first-in-human, SAD Phase 1a study was to evaluate the safety, tolerability, and PK parameters of UV-4 delivered as the hydrochloride salt (**Fig 3**) in healthy subjects, as a necessary precursor to studies in patients with viral disease. In total, 64 subjects were enrolled, 48 of which received UV-4 hydrochloride at one of the eight dose levels (equivalent to 3, 10, 30, 90, 180, 360, 720, or 1000 mg of UV-4) and 16 subjects received placebo (**Fig 1**). The majority of subjects receiving active or placebo were male, 70.8% and 81.3% respectively. Mean age was 27 years (range age 18–44 years). Placebo and active groups were of similar composition for gender, age, race and ethnicity, weight, and body mass index. The demographics of the study participants are presented in **Table 1**.

### Safety

The safety population consisted of all 64 subjects who received investigational product (48 subjects who received active investigational product across eight cohorts, and 16 subjects who

**Table 1. Demographics and Baseline Characteristics.**

| | UV-4 Dose Number in Cohort (N) | | | | | | | | | |
|---|---|---|---|---|---|---|---|---|---|---|
| | 3 mg (N = 6) | 10 mg (N = 6) | 30 mg (N = 6) | 90 mg (N = 6) | 180 mg (N = 6) | 360 mg (N = 6) | 720 mg (N = 6) | 1000 mg (N = 6) | All doses (N = 48) | Placebo (n = 16, [2 per cohort]) |
| Cohort # | 1 | 2 | 3 | 4 | 5 | 6 | 7 | 8 | NA | NA |
| **Sex** | | | | | | | | | | |
| Male | 3 | 4 | 4 | 5 | 5 | 5 | 3 | 5 | 34 | 13 |
| Female | 3 | 2 | 2 | 1 | 1 | 1 | 3 | 1 | 14 | 3 |
| **Race** | | | | | | | | | | |
| White | 6 | 3 | 5 | 4 | 5 | 4 | 4 | 6 | 37 | 14 |
| African American or Black | 0 | 2 | 1 | 2 | 1 | 2 | 2 | 0 | 10 | 2 |
| American Indian or Alaska Native | 0 | 1 | 0 | 0 | 0 | 0 | 0 | 0 | 1 | 0 |
| **Ethnicity** | | | | | | | | | | |
| Hispanic or Latino | 1 | 1 | 0 | 1 | 3 | 0 | 0 | 1 | 7 | 2 |
| Not Hispanic or Latino | 5 | 5 | 6 | 5 | 3 | 6 | 6 | 5 | 41 | 14 |
| **Age (years)** | | | | | | | | | | |
| Mean | 25 | 27 | 27 | 28 | 28 | 29 | 25 | 27 | 27 | 28 |
| Minimum | 21 | 21 | 21 | 19 | 18 | 19 | 18 | 19 | 18 | 18 |
| Maximum | 35 | 41 | 32 | 43 | 43 | 44 | 42 | 38 | 44 | 41 |
| **Weight (kg)** | | | | | | | | | | |
| Mean | 82.9 | 79.6 | 80.8 | 75.9 | 84.0 | 80.1 | 74.1 | 75.0 | 79.1 | 76.2 |
| Minimum | 65.4 | 64.7 | 68.7 | 66.2 | 70.3 | 64.7 | 62.9 | 65.0 | 62.9 | 60.3 |
| Maximum | 99.0 | 102.7 | 95.7 | 88.9 | 94.4 | 95.0 | 95.2 | 82.0 | 102.7 | 95.1 |
| **BMI (kg/m$^2$)** | | | | | | | | | | |
| Mean | 27.44 | 26.88 | 26.44 | 24.76 | 27.05 | 26.13 | 24.86 | 24.93 | 26.06 | 24.89 |
| Minimum | 25.32 | 22.64 | 22.21 | 21.35 | 25.09 | 23.24 | 20.42 | 21.72 | 20.24 | 19.92 |
| Maximum | 28.6 | 29.13 | 28.36 | 29.2 | 29.79 | 28.59 | 28.58 | 26.04 | 29.79 | 28.27 |

received placebo). Safety evaluations were conducted at screening, at admission to the study center, throughout the inpatient portion of the study, and at the follow-up visits.

There were no reported deaths, no SAEs recorded, and no subjects withdrew from the study due to AEs. The majority of treatment-emergent AEs (TEAEs) were mild (Grade 1) or moderate (Grade 2). Mild TEAEs occurred in 85.4% of UV-4 exposed subjects and 75.0% of placebo subjects; and moderate TEAEs in 22.9% of UV-4 exposed subjects and 18.8% of placebo subjects (**Table 2**). Three subjects had severe (Grade 3) TEAEs, including one subject in the 720 mg dose group with prolonged activated partial thromboplastin time (aPTT), increased blood urea and decreased hemoglobin, and two subjects in the placebo group, one with decreased hemoglobin and one with hypoglycemia. Overall, the most commonly reported TEAEs in subjects receiving active investigational product were increased respiratory rate (14/48; 29.2%); proteinuria (10/48; 20.8%); aPTT prolonged, bradycardia, and hypocalcemia (each 9/48; 18.8%); blood urea increased (7/48; 14.6%); and nausea (6/48; 12.5%) (**Table 3**). In the placebo group, the most commonly reported AEs were increased blood urea (4/16; 25.0%), decreased neutrophil count (2/16; 12.5%), increased respiratory rate (2/16, 12.5%), and hypokalemia (each 2/16; 12.5%).

The proportions of subjects reporting at least one related TEAE were 22.9% (11/48) in UV-4 groups versus 18.8% (3/16) in the placebo group (**Table 4**). All related TEAEs were graded mild (Grade 1) to moderate (Grade 2) in severity (**Table 2**). The most commonly reported related TEAE among subjects receiving active was nausea (3/48; 6.3%). At least one related TEAE was reported in three subjects who received 3 mg UV-4, two subjects each who received 10 mg UV-4 and 360 mg UV-4, and one subject each who received 90 mg UV-4 and 720 mg UV-4. No related AEs were reported in subjects who received 30 mg UV-4 and 1000 mg UV-4. All related AEs in subjects who received UV-4 resolved by study end. In the placebo group, three subjects had a total of six TEAEs (each 1/16; 6.3%). Subject 1002 who received placebo had a related TEAE of decreased white blood cell count which was ongoing at study end.

## Safety summary

Halting criteria were met in this study on six occasions. The events triggering study halt were reviewed by the SMC and, in each case, the halt was lifted. Overall, single oral doses of UV-4

**Table 2. Summary of Subjects with Treatment-Emergent Adverse Events by Severity.**

| Adverse Event Severity | | Number of Subjects with TEAE (% of Subjects with TEAE) | | | | | | | | | |
|---|---|---|---|---|---|---|---|---|---|---|---|
| | | UV-4 Dose and Number in Cohort (N) | | | | | | | | | |
| | | 3 mg (N = 6) | 10 mg (N = 6) | 30 mg (N = 6) | 90 mg (N = 6) | 180 mg (N = 6) | 360 mg (N = 6) | 720 mg (N = 6) | 1000 mg (N = 6) | All Doses (N = 48) | Placebo (N = 16) |
| Subjects with ≥1 TEAE | TEAE | 5 (83.3%) | 6 (100.0%) | 5 (83.3%) | 5 (83.3%) | 6 (100.0%) | 5 (83.3%) | 6 (100.0%) | 5 (83.3%) | 43 (89.6%) | 12 (75.0%) |
| | Related TEAE | 3 (50.0%) | 2 (33.3%) | 0 | 1 (16.7%) | 2 (33.3%) | 2 (33.3%) | 1 (16.7%) | 0 | 11 (22.9%) | 3 (18.8%) |
| Grade 1 | TEAE | 5 (83.3%) | 6 (100.0%) | 4 (66.7%) | 5 (83.3%) | 6 (100.0%) | 5 (83.3%) | 5 (83.3%) | 5 (83.3%) | 41 (85.4%) | 12 (75.0%) |
| | Related TEAE | 3 (50.0%) | 2 (33.3%) | 0 | 1 (16.7%) | 1 (16.7%) | 2 (33.3%) | 1 (16.7%) | 0 | 10 (20.8%) | 10 (20.8%) |
| Grade 2 | TEAE | 0 | 1 (16.7%) | 1 (16.7%) | 1 (16.7%) | 2 (33.3%) | 1 (16.7%) | 3 (50.0%) | 2 (33.3%) | 11 (22.9%) | 3 (18.8%) |
| | Related TEAE | 0 | 0 | 0 | 0 | 1 (16.7%) | 0 | 0 | 0 | 1 (2.1%) | 1 (2.1%) |
| Grade 3 | TEAE | 0 | 0 | 0 | 0 | 0 | 0 | 1 (16.7%) | 0 | 1 (2.1%) | 2 (12.5%) |
| | Related TEAE | 0 | 0 | 0 | 0 | 0 | 0 | 0 | 0 | 0 | 0 |

**Table 3. Summary of Subjects with Treatment-Emergent Adverse Events with Incidence Greater Than 5% Among UV-4-Treated Subjects.**

| Preferred Term* | Number of Subjects with TEAE > 5% Incidence (% of Subjects with TEAE > 5% Incidence) | | | | | | | | | |
|---|---|---|---|---|---|---|---|---|---|---|
| | UV-4 Dose and Number in Cohort (N) | | | | | | | | | |
| | 3 mg (N = 6) | 10 mg (N = 6) | 30 mg (N = 6) | 90 mg (N = 6) | 180 mg (N = 6) | 360 mg (N = 6) | 720 mg (N = 6) | 1000 mg (N = 6) | All Doses (N = 48) | Placebo (N = 16) |
| Respiratory rate increased | 3 (50.0) | 4 (66.7) | 1 (16.7) | 2 (33.3) | 1 (16.7) | 1 (16.7) | 1 (16.7) | 1 (16.7) | 14 (29.2) | 2 (12.5) |
| Proteinuria | 2 (33.3) | 1 (16.7) | 1 (16.7) | 0 | 0 | 2 (33.3) | 4 (66.7) | 0 | 10 (20.8) | 0 |
| Activated partial thromboplastin time prolonged | 0 | 3 (50.0) | 0 | 3 (50.0) | 1 (16.7) | 0 | 1 (16.7) | 1 (16.7) | 9 (18.8) | 1 (6.3) |
| Bradycardia | 1 (16.7) | 2 (33.3) | 2 (33.3) | 1 (16.7) | 2 (33.3) | 0 | 1 (16.7) | 0 | 9 (18.8) | 1 (6.3) |
| Hypocalcemia | 0 | 0 | 1 (16.7) | 0 | 2 (33.3) | 2 (33.3) | 3 (50.0) | 1 (16.7) | 9 (18.8) | 1 (6.3) |
| Blood urea increased | 0 | 1 (16.7) | 0 | 2 (33.3) | 1 (16.7) | 1 (16.7) | 1 (16.7) | 1 (16.7) | 7 (14.6) | 4 (25.0) |
| Nausea | 0 | 1 (16.7) | 0 | 0 | 3 (50.0) | 1 (16.7) | 1 (16.7) | 0 | 6 (12.5) | 0 |
| Haematuria | 0 | 0 | 0 | 1 (16.7) | 1 (16.7) | 0 | 2 (33.3) | 0 | 4 (8.3) | 1 (6.3) |
| Prothrombin time prolonged | 0 | 2 (33.3) | 1 (16.7) | 0 | 0 | 1 (16.7) | 0 | 0 | 4 (8.3) | 0 |
| Headache | 1 (16.7) | 0 | 1 (16.7) | 0 | 1 (16.7) | 0 | 0 | 0 | 3 (6.3) | 1 (6.3) |

* Only preferred terms with > 5% incidence among subjects receiving UV-4, sorted by descending frequency. All dose groups were included.

**Table 4. Number of Subjects with Related Treatment-Emergent Adverse Events by System Organ Class and Preferred Term.**

| System Organ Class Preferred Term | Number (%) of Subjects | |
|---|---|---|
| | UV-4 All Doses (N = 48) | Placebo (N = 16) |
| **Total Subjects with ≥ 1 related TEAE** | **11 (22.9%)** | **3 (18.8%)** |
| **Cardiac disorders** | **1 (2.1%)** | **1 (6.3%)** |
| Supraventricular tachycardia | 0 | 1 (6.3%) |
| Ventricular tachycardia | 1 (2.1%) | 0 |
| **Ear and labyrinth disorders** | **1 (2.1%)** | **0** |
| Vertigo | 1 (2.1%) | 0 |
| **Gastrointestinal disorders** | **3 (6.3%)** | **0** |
| Abdominal Pain | 1 (2.1%) | 0 |
| Nausea | 3 (6.3%) | 0 |
| **General disorders and administration site conditions** | **1 (2.1%)** | **0** |
| Local Swelling | 1 (2.1%) | 0 |
| **Investigations** | **2 (4.2%)** | **2 (12.5%)** |
| Blood Calcium Decreased | 1 (2.1%) | 0 |
| Blood Pressure Decreased | 0 | 1 (6.3%) |
| Lipase increased | 1 (2.1%) | 0 |
| Neutrophil count decreased | 0 | 1 (6.3%) |
| White blood cell count decreased | 0 | 1 (6.3%) |
| **Nervous system disorder** | **4 (8.3%)** | **1 (6.3%)** |
| Balance disorder | 1 (2.1%) | 0 |
| Dizziness | 1 (2.1%) | 0 |
| Dizziness postural | 1 (2.1%) | 1 (6.3%) |
| Headache | 2 (4.2%) | 0 |
| Presyncope | 0 | 1 (6.3%) |
| **Renal and urinary disorders** | **1 (2.1%)** | **0** |
| Proteinuria | 1 (2.1%) | 0 |

up to 1000 mg were well tolerated. Dose-dependent increases in TEAEs were not observed. Clinical laboratory results, vital signs, and physical examination data did not reveal any clinically meaningful trends that denoted safety signals throughout the study. During the course of the study, Grade 3 TEAEs (described above) and Grade 2 TEAEs occurring in three or more subjects (hematuria, proteinuria, and decreased hemoglobin) were reviewed by the SRC and SMC and found to be unrelated to study drug. None of the subjects had positive fecal occult blood findings during the study. No dose limiting toxicity was observed; the maximum tolerated dose of UV-4 as a single oral dose in humans has not yet been determined (>1000 mg).

## Pharmacokinetics

All 48 subjects who received UV-4 were included in the PK analysis population. There were no protocol deviations (e.g., incomplete dosing, ingestion of medications, significant deviations in sampling time) or events (e.g., vomiting) potentially affecting PK results. Serial blood and pooled urine samples were collected over a 48-hour post-dose sampling interval for assessment of UV-4 concentrations in plasma and urine.

### UV-4 in Plasma

UV-4 was rapidly absorbed and distributed. Mean UV-4 plasma concentration-time profiles for all active dosing groups are shown in **Fig 4**. A summary of key UV-4 PK parameters is presented in **Table 5** and the full list is described in **S1 Table**. UV-4 concentrations were quantifiable over the entire 48-hour collection interval following oral administration of UV-4 hydrochloride at doses of 10 mg or higher. Following oral administration of the lowest dose (3 mg of UV-4), plasma UV-4 concentrations were quantifiable for 24 hours in 5 of 6 subjects, and for 36 hours in one subject. Across the 3 to 1000 mg UV-4 dose range, median $t_{max}$ ranged from 0.5 hours to 1 hour and appeared to be independent of dose administered.

**UV-4 in Urine.**   UV-4 was quantifiable in pooled urine over the entire 48-hour collection interval for all doses administered (3 to 1000 mg UV-4 dose range). UV-4 recovery in urine ranged from 39.8% to 55.1% across the dose range. The range of individual urinary excretion and mean recovery was lowest for the 3 mg dose. Over the 10 to 1000 mg dose range, urinary recovery ranged from 46.4% to 55.1% and appeared to be independent of dose. Renal clearance (calculated as $A_{e(0-last)}$ divided by $AUC_{(0-last)}$) ranged from 11.1 to 14.5 L/h and was independent of dose. Renal clearance accounted for approximately 50% or greater of UV-4 apparent systemic clearance. The full set of urinary excretion data is included in **S2 Table** and key calculated parameters are in **S1 Table**.

## Assessment of Dose Proportionality

Exposure of UV-4 increased approximately in proportion with dose. Individual and geometric mean $C_{max}$ values versus administered dose are shown in **Fig 5**. Based on the dose proportionality assessment utilizing a power model, the slope was approximately 1.1 when dose proportionality of $C_{max}$ was assessed over both the 333-fold (3–1000 mg) and the 33-fold (30–1000 mg) UV-4 dose range. Similarly, total exposure to UV-4 as measured by $AUC_{(0-inf)}$ increased in a dose-proportional manner over the full dose range (**Fig 6**). A dose-proportionality assessment using a power model calculated a regression slope of 1.013 for $AUC_{(0-inf)}$.

## Discussion

The proposed antiviral mechanism of action of UV-4 is through competitive inhibition of ER α-glucosidases [11,14]. All nascent N-linked glycoproteins, upon synthesis, contain three

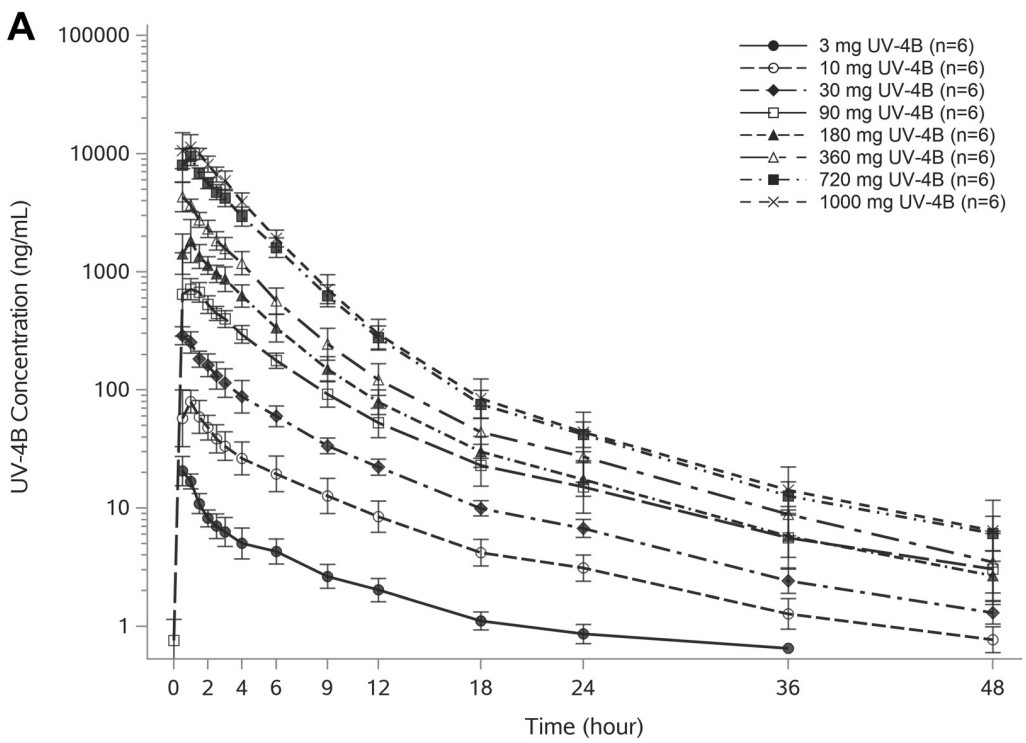

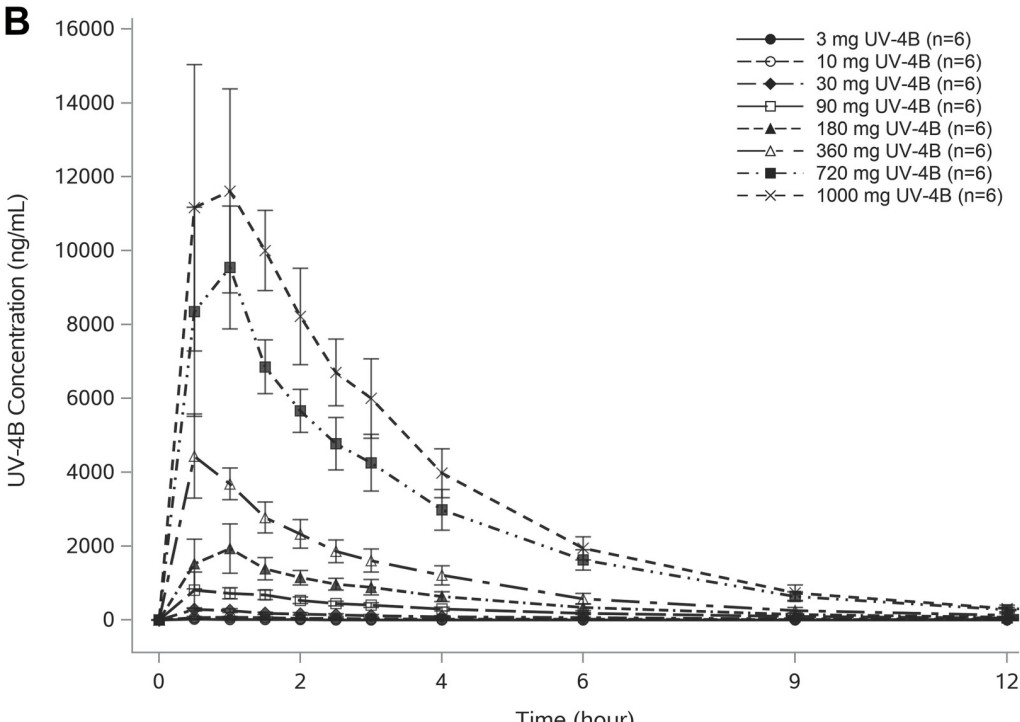

**Fig 4. UV-4 plasma concentration-time profiles. A.** Geometric mean (±SD) UV-4 concentration-time profiles are shown by UV-4 dose on semilogarithmic scale. Absorption was rapid (median $T_{max}$ of 0.5 to 1 hour) and dose independent. Maximum exposure increased approximately in proportion with dose. UV-4 concentrations were quantifiable over the entire 48-hour collection interval following administration of 10 mg or higher dose levels. Following administration of 3 mg UV-4,

UV-4 concentrations were quantifiable for 24 hours in 5 of 6 subjects, and for 36 hours in one subject. **B**. Arithmetic mean (±SD) UV-4 concentration-time profiles are shown by UV-4 dose on linear scale. The graphic represents an expansion of the first 12 hours of sampling to better illustrate UV-4 initial absorption and distribution characteristics.

terminal glucose residues at the distal termini of their N-glycans [35]. Endoplasmic reticulum α-glucosidase I and II are involved in protein folding and assembly via trimming of these terminal glucose residues. Alpha-glucosidase I removes the terminal glucose and α-glucosidase II removes an additional one or two glucoses. After removal of two glucose residues, the glycoprotein becomes a substrate for calnexin and calreticulin, which guide the protein folding process. Inhibition of α-glucosidase I and II leads to viral glycoprotein misfolding and subsequent transport of the misfolded glycoprotein to the proteasome for degradation and elimination [11].

Preclinical toxicology and toxicokinetic studies informed dose selection for the Phase 1a clinical trial. Single and repeat dose studies in mice, rats, and dogs were conducted to evaluate the toxicity and toxicokinetics of UV-4 hydrochloride (detailed safety studies in non-human species will be described in a manuscript in preparation). No observed adverse effect-level (NOAEL) doses were established in all Good Laboratory Practice studies. Based on *in vivo* efficacy studies in AG129 mice following administration of 10 and 20 mg/kg three times daily (TID), UV-4 was predicted to have pharmacological activity at human equivalent dose (HED) as low as 146 to 293 mg/day. Based on the NOAEL for dogs (10 mg/kg/dose TID) and the appropriate conversion recommended in the FDA guidance, the HED for the NOAEL in dog is 333 mg TID, resulting in a 1000 mg dose selected as the highest dose level for this study assuming safety was demonstrated at the lower doses. The starting dose of 3 mg UV-4 was set to one one-hundredth of the HED at the dog NOAEL.

Pharmacokinetic parameters seen in the Phase 1a study were compared to exposure and clearance data collected in animal studies to determine if consistent parameters are observed across species (detailed ADME and PK assessment in non-human species will be described in

**Table 5. Summary of Key UV-4 Pharmacokinetic Parameters.**

| PK Parameters Statistics | UV-4 Dose and Number in Cohort (N) | | | | | | | |
|---|---|---|---|---|---|---|---|---|
|  | 3 mg (N = 6) | 10 mg (N = 6) | 30 mg (N = 6) | 90 mg (N = 6) | 180 mg (N = 6) | 360 mg (N = 6) | 720 mg (N = 6) | 1000 mg (N = 6) |
| $AUC_{(0-inf)}$ (h*ng/mL) |  |  |  |  |  |  |  |  |
| Arithmetic Mean | 95.6 | 431 | 1261.7 | 3728.3 | 7110 | 13833.3 | 32916.7 | 43083.3 |
| CV (%) | 16.3 | 26.7 | 9.6 | 13.8 | 8.1 | 14.1 | 10.7 | 13.5 |
| Geometric Mean | 94.6 | 420.1 | 1256.4 | 3700.5 | 7090.4 | 13714.7 | 32756.2 | 42765.2 |
| Cmax (ng/mL) |  |  |  |  |  |  |  |  |
| Arithmetic Mean | 22.4 | 84.6 | 292.5 | 945.5 | 2148.3 | 4568.3 | 9925 | 13133.3 |
| CV (%) | 15.9 | 23.4 | 15.3 | 34.9 | 28.8 | 19.7 | 20.7 | 17.1 |
| Geometric Mean | 22.1 | 82.6 | 289.1 | 899.8 | 2062.6 | 4494.6 | 9758.9 | 12993.2 |
| Half-Life (h) |  |  |  |  |  |  |  |  |
| Arithmetic Mean | 10.7 | 12 | 10 | 10.4 | 8.7 | 8.1 | 8.2 | 8.8 |
| CV (%) | 33.2 | 10.4 | 7.6 | 27.1 | 11.2 | 9.7 | 6.9 | 27.9 |
| Geometric Mean | 10.3 | 11.9 | 10 | 10.2 | 8.7 | 8.1 | 8.2 | 8.6 |
| Tmax (h) |  |  |  |  |  |  |  |  |
| Median | 0.5 | 1 | 0.5 | 0.5 | 0.8 | 0.5 | 0.8 | 1 |
| Minimum | 0.5 | 0.5 | 0.5 | 0.5 | 0.5 | 0.5 | 0.5 | 0.5 |
| Maximum | 1 | 1.5 | 1 | 1.5 | 1 | 1 | 1 | 1.5 |

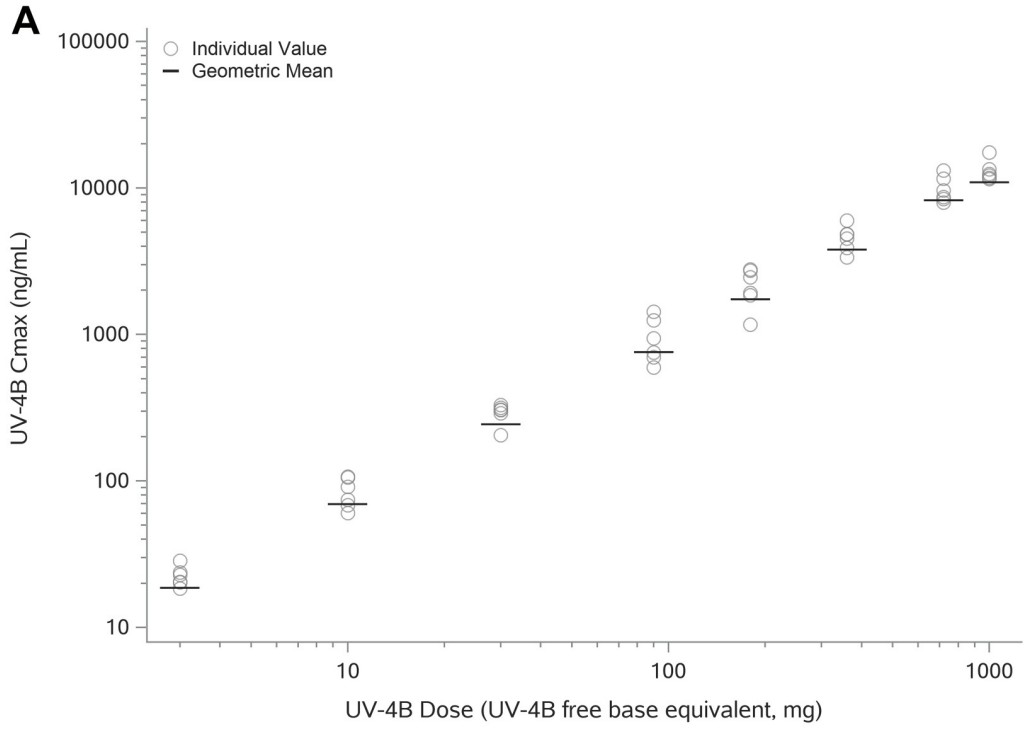

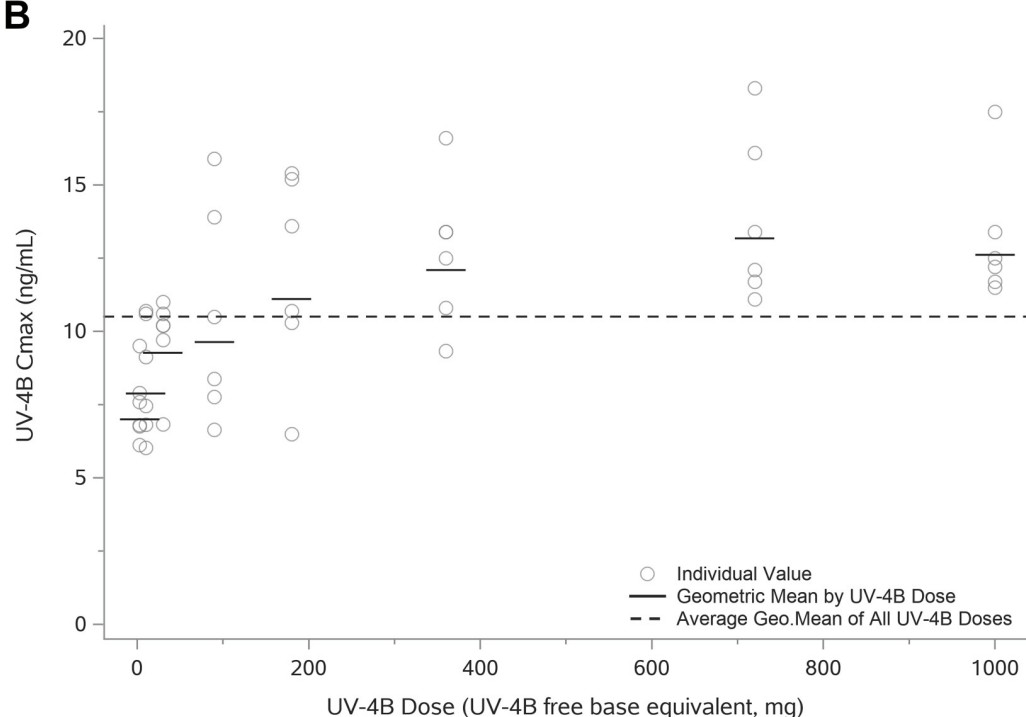

**Fig 5. Individual and Geometric Mean of $C_{max}$ versus Administered Dose. A**. Individual and geometric mean of $C_{max}$ represented using log-scale. Overall $C_{max}$ appears to increase in a dose proportional manner. **B**. Individual and geometric mean of Dose Normalized $C_{max}$ (DNC$_{max}$) (linear scale). There is considerable overlap in the range of individual DNC$_{max}$ values, with a trend of a slightly greater than proportional increase in $C_{max}$ over the 3 to 180 mg UV-4 dose range. The range and mean DNC$_{max}$ values were similar across the 360 to 1000 mg dose range. Dose proportionality was assessed utilizing a power model, the slope was approximately 1.1 over either the 333-fold (3 to 1000 mg) or the 33-fold (30 to 1000 mg) UV-4

dose range. The 90% CI estimates of 333-fold (1.078, 1.133) and 33-fold (1.048, 1.157) were not fully contained within the pre-specified confidence bounds (0.94, 1.06). Overall $C_{max}$ appears to increase in a generally dose proportional manner, especially at the higher doses of 360 mg and above.

a manuscript in preparation). Nonclinical PK and excretion studies of UV-4 as the hydrochloride salt and UV-4 free base were conducted using the oral and IV routes of administration as single doses in mice, rats, ferrets, and dogs. Human systemic clearance was estimated at 33.6 L/h based on simple scaling of systemic clearance values against body weight for the IV PK data obtained in mouse, rat, and dog. Using this estimate, total exposure ($AUC_{(0- inf)}$) following a starting dose of 3 mg was predicted to be 89 ng·h/mL, compared to 95.6 ng·h/mL observed in this study. Absorption was rapid following oral administration in each nonclinical species, in alignment with the median $T_{max}$ of 0.5 to 1 hour across all dose groups in the Phase 1a study. Elimination is also rapid in all species studied with approximate half-life ($t_{1/2}$) values 2 to 9 hours in ferrets, mice, rats, and dogs, comparable to mean half-life ranging from 8.14 to 12.0 hours in the Phase 1a study. Urinary excretion of UV-4 after an oral dose of UV-4 hydrochloride in male Sprague Dawley rats resulted in 45% of dose recovered in urine over a period of 48 hours, compared to urinary recovery ranging from 46.4% to 55.1% in 48 hours over the 10 to 1000 mg dose range in the Phase 1a study. These comparisons confirm that the PK characteristics of UV-4 hydrochloride are generally consistent between nonclinical species and humans, which supports likelihood that the *in vivo* antiviral efficacy demonstrated for UV-4 and UV-4B in multiple animal models may translate into humans.

The data from this study demonstrate consistent and predictable pharmacokinetics across the wide dose range studied (333-fold, 3mg to 1000mg). UV-4 concentration values over time for each dose were similar within cohorts (low standard deviation), particularly given that UV-4 was administered orally and on a fixed-dose basis. $C_{max}$ values and total exposure (plotted for each dose) exhibited linearity over much of the range (**Figs 5** and **6**). This linearity in PK behavior, and the lack of significant toxicity findings even at the highest dose tested, suggests that higher exposure to UV-4 may be well tolerated, however this would need to be evaluated in a formal expansion to this Phase 1a study.

There are five iminosugar compounds currently approved for use in humans. These are miglustat (Zavesca), migalastat (Galafold), miglitol (Glyset), acarbose (Precose), and (ex-US) voglibose (Basen). In addition, celgosivir (a pro-drug of castanospermine) has been tested in several clinical studies and is currently in development for dengue virus infection [36]. Together, these therapies provide a significant safety database for the iminosugar class of compounds. Relevant safety findings of these iminosugars are discussed briefly below with correlations to the observations from this Phase 1a study.

Several iminosugars approved for human use may cause gastrointestinal (GI) effects including diarrhea, flatulence, and abdominal bloating and discomfort as indicated for examples on the package inserts for Zavesca and Glyset. The cause of these symptoms is believed to be due to the inhibition of GI-resident disaccharidases which leads to an osmotic diarrhea. Diarrhea is most severe in the early days of dosing and thus is relevant to the acute use scenario that would typify use of an iminosugar in Dengue and other acute viral diseases. In a dog dose-range finding study, animals receiving the highest dose (150 mg UV-4/kg/dose TID) for seven days had severe GI effects, and at doses of 10 mg/kg/dose TID diarrhea was still apparent although not adverse; similar findings were observed in a 14-day dog study. However, no incidence of Bristol Stool Scale Type 7 [37] occurred in this Phase 1a study. Most of the subjects had a stool frequency of once or twice per day, except for Subject 5008 (180 mg UV-4), who had a stool frequency of thrice per day on Day 1 of the study. None of the subjects had positive

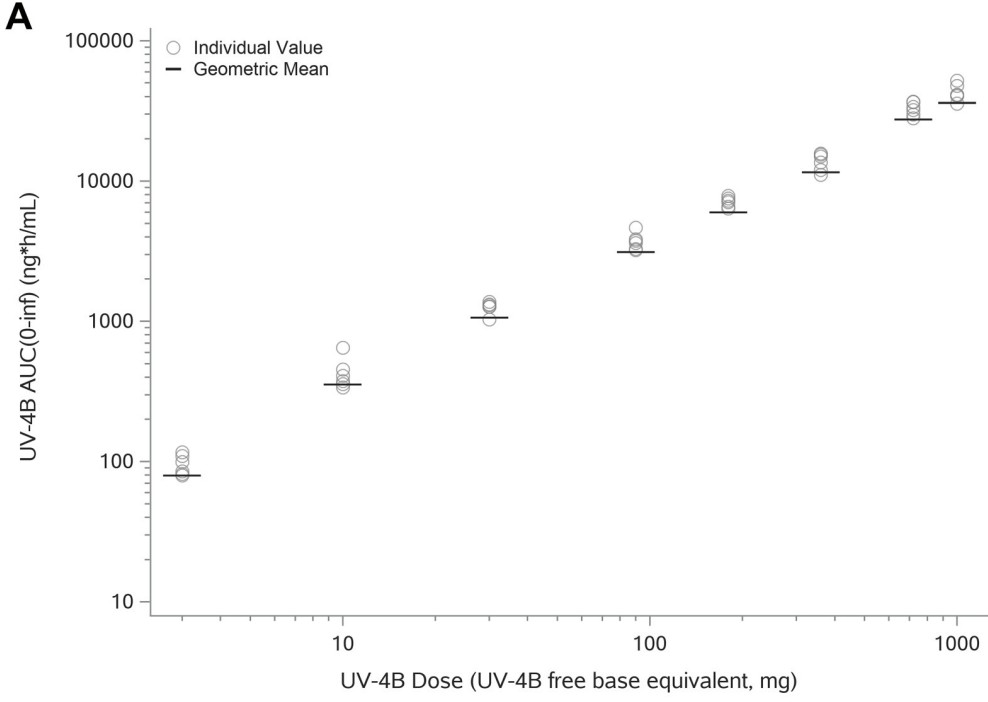

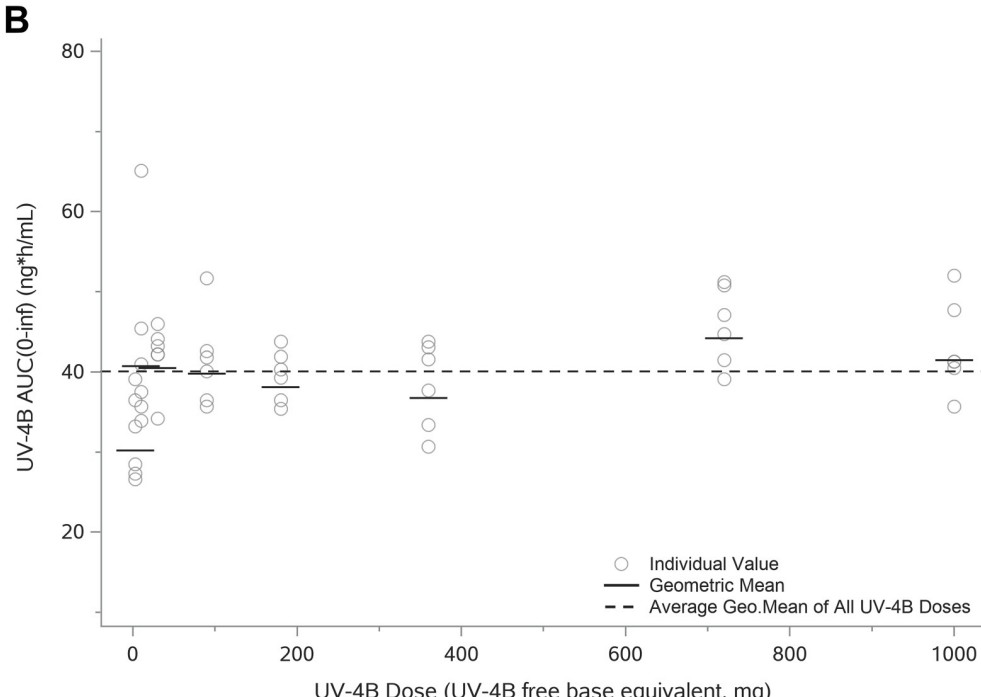

**Fig 6. Individual and Geometric Mean AUC$_{(0\text{-inf})}$ versus Administered Dose. A**. Individual and geometric mean of AUC$_{(0\text{-inf})}$ represented using log-scale. Overall AUC$_{(0\text{-inf})}$ appears to increase in a generally dose proportional manner. **B**. Individual and geometric mean Dose Normalized AUC$_{(0\text{-inf})}$ (DNAUC$_{(0\text{-inf})}$) (linear scale). AUC$_{(0\text{-inf})}$ increased in a dose-proportional manner over the 3 to 1000 mg (333-fold) UV-4 dose range. Mean'and individual range of DNAUC$_{(0\text{-inf})}$ of 3 mg group was slightly lower compared to the subsequent dose levels (10 to 1000 mg), for which the individual range and mean DNAUC$_{(0\text{-inf})}$ values were similar. Dose proportionality was assessed utilizing a power model. The slope was 1.033 when dose proportionality was assessed over the 333-fold dose range, but the 90% CI estimates (1.013, 1.053) were not fully contained within the pre-specified confidence bounds (0.96, 1.04). When the dose proportionality assessment was repeated over a

33-fold dose range (30 to 1000 mg), the slope was 1.013 and the 90% CI estimates (0.9837, 1.042) were fully contained within the pre-specified confidence bounds (0.94, 1.06). These data indicate that increases of $AUC_{(0-inf)}$ were dose proportional over the 30 to 1000 mg dose range.

fecal occult findings during the study. Other GI findings were mild and infrequent. Among the 48 subjects who received UV-4, two reported abdominal pain, six reported nausea, and one vomited. There was no obvious dose proportionality to these observations; incidence was highest in the 180 mg cohort, while no subjects in the 1000 mg cohort experienced GI symptoms.

In one controlled trial of 212 obese patients treated for 36 months with the structurally-unrelated iminosugar acarbose, alanine aminotransferase (ALT) elevations occurred in 9% of acarbose vs. 2% of placebo controls; all abnormal ALT values resolved rapidly with discontinuation of drug [38]. A review focused on safety of 1108 patients treated with acarbose for an average of 6 months demonstrated that rates of ALT elevations were no different between acarbose- and placebo-treated patients [39]. Miglustat does not appear to be associated with aspartate aminotransferase (AST) and ALT elevations in humans with Gaucher disease, Niemann-Pick disease (NPC), and Fabry disease (Zavesca SmPC), at doses of 100 mg TID or less [40]. Celgosivir, a pro-drug α-glucosidase I inhibitor was assessed for safety and tolerability at doses of 300 mg BID; the study was prematurely discontinued due to grade 3 transaminase, creatine phosphokinase (CK), and lactate dehydrogenase (LDH) elevations. During the preclinical studies in dogs and rats, exposure to UV-4 increased AST and ALT and produced increased AST/ALT ratio. In dogs, this AST/ALT ratio change was seen after a single dose. A NOEL for the AST increases has not been identified in dogs. At this time, the source of increased AST and ALT in nonclinical animals is unclear; there was no evidence of impairment of hepatic function, and no histopathologic changes were observed in the liver. Similarly, CK remained normal and there was no histologic evidence of muscle damage. Importantly, there was no corollary to these non-clinical findings in this study. No human subjects receiving UV-4 as a single dose up to 1000 mg experienced an increase in AST. While two subjects in this study had an increase in ALT, these were not dose-related in incidence (one subject at 180 mg, and one subject at 1000 mg; and no subjects at 360 mg or 720 mg doses).

Animal studies have raised concerns regarding the neurologic safety of miglustat (https://www.accessdata.fda.gov/drugsatfda_docs/label/2010/021348s008lbl.pdf). These findings have not been supported by long-term miglustat safety data in humans [40]. In the preclinical evaluations of UV-4 hydrochloride, tremors were seen in beagle dogs receiving a single dose of 400 mg UV-4/kg. In this Phase 1a study of UV-4 hydrochloride, the nervous system AEs observed were balance disorder (1/48 active subjects), dizziness (1/48 active subjects), dizziness postural (1/48 active subjects, 1/16 placebo subjects), headache (2/48 active subjects), and presyncope (1/16 placebo subjects), and the incidence was not dose-related.

Published data from numerous laboratories have confirmed the broad-spectrum antiviral activity of iminosugars such as UV-4 on filovirus [15,41], Flaviviridae [42–44], and influenza virus [13,27]. This broad-spectrum activity has been extended recently to the coronaviruses including SARS-COV2 [29,45] Franco et al., submitted for publication]. UV-4 has also recently been shown to have beneficial effects on the host's inflammatory response which is a hallmark of dengue, COVID-19, and other infectious diseases [31]. Dengue viruses cause pathophysiological responses that can respond to treatment. Dengue virus non-structural protein, NS1, is a direct endothelial toxin potentially responsible for the leakage of fluid from the circulatory system that may lead to dengue shock syndrome and death [18,46,47]. An antiviral compound may or may not restore endothelial integrity. It is expected that pathophysiological

responses would be reduced by action of an active antiviral drug administered soon after infection, but this would require clinical evaluation in later studies in infected patients. Our studies have confirmed with both dengue virus and influenza the low potential for development of viral resistance against this host-targeted therapeutic approach [26,28]. We also recently published a novel high-intensity dosing approach for UV-4, using one or several individual doses given at intervals which would not allow for accumulation in plasma [48]. The pharmacodynamic effect of a single dose of UV-4 on the target ER α-glucosidase enzymes extends for at least 72 hours, well beyond the time of significant plasma exposure. The safety and PK data from this Phase 1a study support further evaluation of this high-intensity dosing approach in a subsequent study to provide a modest 2x-3x higher dose (to match efficacious doses in mouse models) and/or evaluate several individual doses.

Notwithstanding the well-known limitations of a Phase 1a study related to sample size, in this randomized double-blind study single oral doses of UV-4 hydrochloride up to and including 1000 mg of UV-4 were safe and well tolerated in human subjects. There were no SAEs reported and no subjects withdrawn from the study due to AEs. No dose-dependent increases of AEs were observed. Pharmacokinetics data indicate predictable and dose-dependent exposures across the 333-fold dose range evaluated. It has been hypothesized that host-targeted therapeutics may have increased toxicity risk due to disruption of cellular pathways in both infected and healthy cells [49], but the data here indicate that such a hypothesis must be evaluated on a case-by-case basis. The results from this study, and the overall safety profile for the five iminosugars approved for clinical use, support the development of iminosugars including UV-4 as potential host-targeted antiviral therapies.

## Supporting information

**S1 Table. Summary of Key UV-4 Pharmacokinetic Parameters.**
(DOCX)

**S2 Table. UV-4 Urinary Recovery.**
(DOCX)

**S3 Table. Concentrations of UV-4 in Human Plasma Samples.**
(DOC)

**S1 File. Clinical Study Protocol.**
(PDF)

**S2 File. CONSORT Checklist for Reporting a Randomized Trial.**
(PDF)

**S3 File. Data Availability Statement.**
(DOCX)

## Acknowledgments

We thank Lisa Beth Ferstenberg for assistance with development of the clinical program, Abhinava Reddy for transferring clinical data into Emergent, Chandra Nimbal and Mona Sharma for data analysis, Michael Li for assistance with statistics and review of the manuscript, and Annie Jones and Christine Hall for manuscript review.

## Author Contributions

**Conceptualization:** Michael Callahan, Marla Smith, Matthew Duchars, Aruna Sampath, Urban Ramstedt.

**Data curation:** Grace Lin, Brian Kaufman.

**Formal analysis:** Anthony M. Treston, Grace Lin, Lisa Evans DeWald, Kevin Spurgers.

**Investigation:** Michael Callahan, Anthony M. Treston, Marla Smith, Brian Kaufman, Preeya Lowe, Matthew Duchars, Aruna Sampath, Urban Ramstedt.

**Resources:** Mansoora Khaliq, Kelly L. Warfield.

**Writing – original draft:** Anthony M. Treston, Lisa Evans DeWald, Kevin Spurgers, Kelly L. Warfield.

**Writing – review & editing:** Anthony M. Treston, Lisa Evans DeWald, Kevin Spurgers, Kelly L. Warfield.

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
