## [Decision Letter · Decision Letter 0]

17 Feb 2022

Dear Dr. Warfield,

Thank you very much for submitting your manuscript "Randomized Single Oral Dose Phase 1 Study of Safety, Tolerability, and Pharmacokinetics of Iminosugar UV-4 Hydrochloride (UV-4B) in Healthy Subjects" for consideration at PLOS Neglected Tropical Diseases. As with all papers reviewed by the journal, your manuscript was reviewed by members of the editorial board and by at least 1 independent reviewer. In light of the feedback (below this email), we would like to invite the resubmission of a revised version that takes into account the reviewer's comments. 

We cannot make any decision about publication until we have seen the revised manuscript and your response to the reviewers' comments. Your revised manuscript is also likely to be sent to reviewers for further evaluation.

Sincerely,

Robert C Reiner, Ph. D

Deputy Editor

Reviewer's Responses to Questions

**Key Review Criteria Required for Acceptance?**

**Methods**

-Are the objectives of the study clearly articulated with a clear testable hypothesis stated?

-Is the study design appropriate to address the stated objectives?

-Is the population clearly described and appropriate for the hypothesis being tested?

-Is the sample size sufficient to ensure adequate power to address the hypothesis being tested?

-Were correct statistical analysis used to support conclusions?

-Are there concerns about ethical or regulatory requirements being met?

Reviewer #1: As a phase 1a study of a potential antiviral compound, this is well designed and conducted and meets ethical and regulatory requirements.

**Results**

-Does the analysis presented match the analysis plan?

-Are the results clearly and completely presented?

-Are the figures (Tables, Images) of sufficient quality for clarity?

Reviewer #1: This is a phase 1a clinical trial of an ascending single oral dose in healthy adults of hydrochloride salt equivalent to 3, 10, 30, 90, 180, 360, 720, or 1000 mg of UV-4 ( N -(9’-methoxynonyl)-1-deoxynojirimycinas (6 subjects per cohort), or placebo (2 subjects per cohort).The study was designed to evaluate the safety, tolerability, and pharmacokinetics of this compound. The dose range tested was not justified by any published evidence of the bio-efficacy of this compound against dengue viruses in any animal model. This fact should be acknowledged and explained in the Introduction and Discussion. The authors should drastically shorten the introduction which should clearly state that this compound has been shown to have antiviral properties in vitro and is expected to exert its value in vivo in humans as an antiviral compound. The authors should understand and convey clearly to readers that in humans, dengue viruses cause pathophysiological responses that do respond to treatment. Dengue virus non-structural protein, NS1, is a direct endothelial toxin responsible for the potentially lethal leakage of fluid from the circulatory system that may lead to dengue shock syndrome and death. The use of the word “treatment” in dengue should be reserved for efforts that restore fluid volume or reverse endothelial damage. An antiviral compound may or may not restore endothelial integrity. Careful clinical evaluations will be required to establish this as a therapeutic outcome. Throughout this manuscript, the pharmacological action of UV-4B should be referred to as “antiviral.” 

Specific comments: 

243. Remove the word “designed” as it is redundant.

**Conclusions**

-Are the conclusions supported by the data presented?

-Are the limitations of analysis clearly described?

-Do the authors discuss how these data can be helpful to advance our understanding of the topic under study?

-Is public health relevance addressed?

Reviewer #1: As stated above, this compound should not be referred to as a "treatment" of potentially lethal human dengue virus infections. It is difficult for the reviewer to understand how the authors selected the dosage range for testing with no efficacy data available from an animal model.

**Editorial and Data Presentation Modifications?**

Reviewer #1: Several important suggestions have been made above. Satisfactory responses are crucial to the publication of an acceptable phase 1a study.
---

## [Decision Letter · Decision Letter 1]

10 Jun 2022

Dear Dr. Warfield,

Thank you very much for submitting your manuscript "Randomized Single Oral Dose Phase 1 Study of Safety, Tolerability, and Pharmacokinetics of Iminosugar UV-4 Hydrochloride (UV-4B) in Healthy Subjects" for consideration at PLOS Neglected Tropical Diseases. As with all papers reviewed by the journal, your manuscript was reviewed by members of the editorial board and by several independent reviewers. The reviewers appreciated the attention to an important topic. Based on the reviews, we are likely to accept this manuscript for publication, providing that you modify the manuscript according to the review recommendations. 

Sincerely,

Robert C Reiner, Ph. D

Deputy Editor

Reviewer's Responses to Questions

**Key Review Criteria Required for Acceptance?**

**Methods**

-Are the objectives of the study clearly articulated with a clear testable hypothesis stated?

-Is the study design appropriate to address the stated objectives?

-Is the population clearly described and appropriate for the hypothesis being tested?

-Is the sample size sufficient to ensure adequate power to address the hypothesis being tested?

-Were correct statistical analysis used to support conclusions?

-Are there concerns about ethical or regulatory requirements being met?

Reviewer #1: Excellent

**Results**

-Does the analysis presented match the analysis plan?

-Are the results clearly and completely presented?

-Are the figures (Tables, Images) of sufficient quality for clarity?

Reviewer #1: This is a large phase 1 evaluation of the safety, tolerability and pharmokinetics of a therapeutic immunosugar involving 64 adult subjects who received single oral doses of UV-4 as the hydrochloride salt equivalent to 3, 10, 30, 90, 180, 360, 720, or 1000 mg (6 subjects per cohort), or placebo (2 subjects per cohort). In this well designed and executed study all single doses of UV-4 hydrochloride were well tolerated with no serious adverse events or dose-dependent increases in adverse events observed. References 17 – 20 describing current concepts of dengue epidemiology, pathogenesis, dengue vaccine efficacy and safety could be more authoritative and accessible. (1-5)

1. Sridhar S, Luedtke A, Langevin E, Zhu M, Bonaparte M, Machabert T, et al. Effect of Dengue Serostatus on Dengue Vaccine Safety and Efficacy. N Engl J Med. 2018;379(4):327-40.

2. Glasner DR, Puerta-Guardo H, Beatty PR, Harris E. The Good, the Bad, and the Shocking: The Multiple Roles of Dengue Virus Nonstructural Protein 1 in Protection and Pathogenesis. Annual review of virology. 2018;5(1):227-53.

3. Halstead SB, Russell PK, Brandt WE. NS1, Dengue's Dagger. J Infect Dis. 2020;221:857-60.

4. Halstead SB. Is dengue vaccine protection possible? Clinical Infectious Diseases. 2021;72(https://doi.org/10.1093/cid/ciab282).

5. Halstead SB. Vaccine-Associated Enhanced Viral Disease: Implications for Viral Vaccine Development. BioDrugs : clinical immunotherapeutics, biopharmaceuticals and gene therapy. 2021;35(5):505-15.

**Conclusions**

-Are the conclusions supported by the data presented?

-Are the limitations of analysis clearly described?

-Do the authors discuss how these data can be helpful to advance our understanding of the topic under study?

-Is public health relevance addressed?

Reviewer #1: Excellent

**Editorial and Data Presentation Modifications?**

Reviewer #1: None

**Summary and General Comments**

Reviewer #1: Suggested additional useful references for consideration.

PLOS authors have the option to publish the peer review history of their article (what does this mean?). If published, this will include your full peer review and any attached files.

Reviewer #1: No

Figure Files:

Data Requirements:

Reproducibility:

References

---

## [Editor Report · Decision Letter 2]

5 Jul 2022

Dear Dr. Warfield,

We are pleased to inform you that your manuscript 'Randomized Single Oral Dose Phase 1 Study of Safety, Tolerability, and Pharmacokinetics of Iminosugar UV-4 Hydrochloride (UV-4B) in Healthy Subjects' has been provisionally accepted for publication in PLOS Neglected Tropical Diseases.

Best regards,

Robert C Reiner, Ph. D

Deputy Editor

---

## [Editor Report · Acceptance letter]

4 Aug 2022

Dear Dr. Warfield,

We are delighted to inform you that your manuscript, "Randomized Single Oral Dose Phase 1 Study of Safety, Tolerability, and Pharmacokinetics of Iminosugar UV-4 Hydrochloride (UV-4B) in Healthy Subjects," has been formally accepted for publication in PLOS Neglected Tropical Diseases.

Best regards,

Shaden Kamhawi

co-Editor-in-Chief

Paul Brindley

co-Editor-in-Chief
